# High-throughput widefield fluorescence imaging of 3D samples using deep learning for 2D projection image restoration

Edvin Forsgren[1][◎]*, Christoffer Edlund[2][◎], Miniver Oliver[3][¤], Kalpana Barnes[3], Rickard Sjögren[2], Timothy R. Jackson[3]

**1** Computational Life Science Cluster (CLiC), Department of Chemistry, Umeå University, Umeå, Sweden, **2** Sartorius Corporate Research, Sartorius Stedim Data Analytics AB, Umeå, Sweden, **3** Sartorius BioAnalytics, Essen BioScience, Ltd., Units 2 & 3 The Quadrant, Royston, Hertfordshire, United Kingdom

◎ These authors contributed equally to this work.
¤ Current address: Biopharm Discovery, GlaxoSmithKline, Stevenage, Hertfordshire, United Kingdom
* edvin.forsgren@umu.se

**Data Availability Statement:** The data used in the article is publicly available at https://figshare.com/projects/Dataset_of_fluorecent_3D-samples_projected_to_2D/126629 as well as the source

## Abstract

Fluorescence microscopy is a core method for visualizing and quantifying the spatial and temporal dynamics of complex biological processes. While many fluorescent microscopy techniques exist, due to its cost-effectiveness and accessibility, widefield fluorescent imaging remains one of the most widely used. To accomplish imaging of 3D samples, conventional widefield fluorescence imaging entails acquiring a sequence of 2D images spaced along the z-dimension, typically called a z-stack. Oftentimes, the first step in an analysis pipeline is to project that 3D volume into a single 2D image because 3D image data can be cumbersome to manage and challenging to analyze and interpret. Furthermore, z-stack acquisition is often time-consuming, which consequently may induce photodamage to the biological sample; these are major barriers for workflows that require high-throughput, such as drug screening. As an alternative to z-stacks, axial sweep acquisition schemes have been proposed to circumvent these drawbacks and offer potential of 100-fold faster image acquisition for 3D-samples compared to z-stack acquisition. Unfortunately, these acquisition techniques generate low-quality 2D z-projected images that require restoration with unwieldy, computationally heavy algorithms before the images can be interrogated. We propose a novel workflow to combine axial z-sweep acquisition with deep learning-based image restoration, ultimately enabling high-throughput and high-quality imaging of complex 3D-samples using 2D projection images. To demonstrate the capabilities of our proposed workflow, we apply it to live-cell imaging of large 3D tumor spheroid cultures and find we can produce high-fidelity images appropriate for quantitative analysis. Therefore, we conclude that combining axial z-sweep image acquisition with deep learning-based image restoration enables high-throughput and high-quality fluorescence imaging of complex 3D biological samples.

code implementing our methods at https://github.com/edvinforsgren/ProjSweep.

**Funding:** EF is partly founded by Swedish National Strategic e-Science Research Program eSSENCE. The funders had no role in study design, data collection and analysis, decision to publish, or preparation of the manuscript.

**Competing interests:** The authors have declared that no competing interests exist.

## Introduction

Fluorescence microscopy is an indispensable tool to study biological phenomena, having enabled countless discoveries over the past several decades. For live-cell imaging in particular, fluorescence microscopy enables interrogation of the spatial and temporal dynamics of complex biological processes. By using fluorescent proteins (FPs) to label endogenous cell signaling molecules, one can gain insight into the heterogeneity and kinetics of changes to cellular processes over time through fluorescence intensity measurements. However, fluorescence imaging has inherent challenges and requires careful balancing of light exposure to samples and overall acquisition time with the signal-to-noise ratio [1, 2]. Overexposure of samples can cause FP photobleaching or phototoxicity, while underexposure will generate noise-dominated images. Oftentimes, to obtain meaningful data, acquisition parameters must be uniquely tuned to the specific FP and biological sample.

These matters are further complicated when considering three-dimensional (3D) samples. For the most widely used modalities of fluorescent imaging, i.e. widefield and confocal, conventional 3D fluorescent imaging entails acquiring a step-wise sequence of two-dimensional (2D) optical slices spaced across the axial dimension, which is commonly referred to as a z-stack [2]. Optimal axial sampling rates are determined by the z-axis resolution of the microscope [3] but are usually impractical for large samples, which may require images at hundreds of z-locations. But because many applications do not require axial information and 3D data can become cumbersome to manage, much of the time it is sufficient to project 3D z-stacks into a single 2D image [4]. Many commercially available and open-source software packages exist that enable this feature of producing z-projected images [5, 6].

Despite technical challenges, recent trends in cellular research continues to see focus shift from 2D monolayers of cells toward more physiologically relevant three-dimensional 3D models, including tumor spheroids and organoids [7, 8]. 3D models facilitate greater insight into microenvironmental factors and spatial interactions, incorporating elements that are better representative of conditions seen in vivo. While imaging techniques such as lattice light sheet fluorescence microscopy promise high-throughput, high-quality fluorescent imaging of such samples [9], they require sophisticated hardware and user expertise. To increase the throughput for widefield fluorescent imaging, an alternative acquisition scheme utilizing axial sweeping was proposed, which, in combination with light modulation and point-spread function (PSF)-based image restoration, could generate an axially compressed 2D image of a 3D sample [10]. While this strategy was promising in its 2- to 10-fold acquisition speed increase, its hardware requirements and complex traditional computer vision-based approach limits its practicality and implementation to complex biological samples. More recently, an acquisition scheme was proposed utilizing sweeping synchronized galvanometric mirrors in a multi-angle projection system to accelerate imaging speeds by a factor of $> 100$ [4]. While indeed promising, this technique requires a specialized hardware setup and was only tested in confocal and light-sheet fluorescence microscopy, making its applicability unclear for widefield fluorescence imaging.

Whereas traditional PSF-based image restoration is incredibly challenging for complex 3D samples, convolutional neural networks (CNNs) are now routinely used for such difficult computer vision problems. Across various 3D fluorescence imaging studies, CNNs are used to segment nuclei [11, 12] and cell bodies [13, 14], image restoration [15, 16], image super-resolution [17, 18], as well as blind deconvolution in 2D widefield fluorescence imaging [19, 20]. In the 3D fluorescence imaging space, recent studies have outlined methods to use CNNs for virtual refocusing to predict a user-defined 3D surface from a single plane 2D fluorescent image [21, 22], using recurrent neural networks (RNNs) to reconstruct a complete 3D volume

with far fewer optical sections than typically needed [23], and optimization of phase mask filters for PSF-engineering to capture 2D extended depth-of-field images or 3D volumetric images [24, 25]. While these notable advancements demonstrate the potential of CNNs to increase throughput for 3D fluorescent imaging, they have primarily been limited to high-resolution, subcellular imaging. These studies do not address the need for improving throughput capabilities for imaging large 3D models, such as tumor spheroids and organoids, which can be larger than a millimeter in diameter. Therefore, there remains a gap in the field to address high-throughput fluorescent imaging of large, complex 3D samples without complicated hardware or acquiring a large number of z-slices.

In this paper we present a novel workflow to generate high-quality 2D projection images of entire 3D fluorescent biological specimens while minimizing exposure times and maximizing throughput (summarized in Fig 1). Using live 3D tumor spheroids, we collected a dataset comprising image pairs of conventional projected z-stacks (Fig 1a) and the corresponding images formed from a fast axial sweep acquisition, which we are terming a z-sweep image (Fig 1b). We then train CNNs on the paired data, in order to achieve a model that can faithfully reconstruct high-quality 2D projections of z-stacks from the comparatively low-quality z-sweep images (Fig 1c). By combining state-of-the-art deep learning-based computer vision with high-throughput image acquisition, we enable low-phototoxicity, real-time quantitative analysis of live 3D samples using widefield fluorescence imaging.

## Methodology

Formally, we aim to map a z-sweep image, $S \in \mathbb{R}^{h \times w}$, to the corresponding z-projection image, $P \in \mathbb{R}^{h \times w}$, by means of a generator $G: \mathbb{R}^{h \times w} \to \mathbb{R}^{h \times w}$ such that $G(S) = \hat{P} \approx P$, where $G$ is a CNN parameterized by $\theta$. Given a dataset of $N$ pairs of z-sweep and z-projection images, we

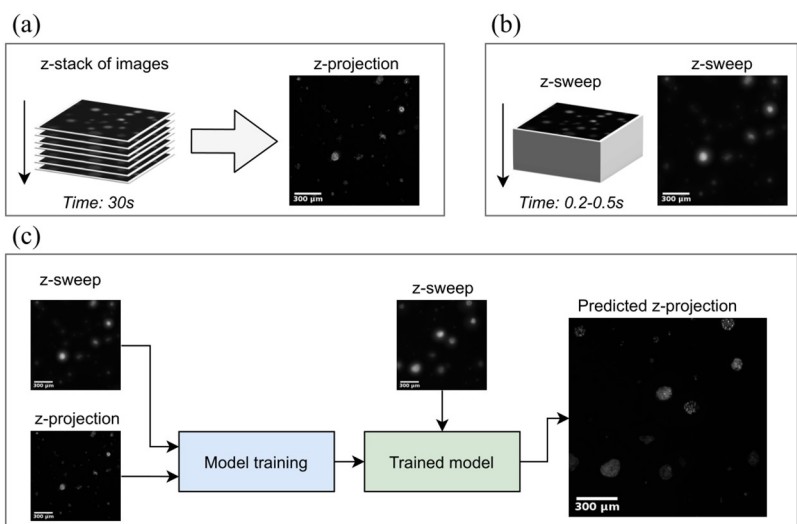

**Fig 1. Schematic overview of the proposed workflow.** Paired data comprising images from two different acquisition paradigms were collecting, z-stack projection and z-sweep images. (a) Z-projection images are generated by first collecting a stack of fluorescent images at discrete steps along the z-direction and projecting them down to a high quality 2D-image. (b) Z-sweeps are acquired with a single or consecutive exposures while the camera continuously moves through the axial dimension, yielding a single 2D image with intensity integrated over the z-axis. (c) Paired data of z-sweeps and z-projections are used to train a CNN, which is then capable of predicting the high-quality z-projection images from z-sweep images, to enable high-throughput acquisition of high-quality images.

train $G$ to minimize the loss function $L : \mathbb{R}^{h \times w} \times \mathbb{R}^{h \times w} \to \mathbb{R}$ to find optimal parameters:

$$\hat{\theta} = \arg \min_{\theta} \sum_{i=1}^{N} L(G(S_i \mid \theta), P_i). \tag{1}$$

The sections that follow will describe the CNN architectures investigated in this paper and the loss functions used to train them.

## Image generator

In this work, all generators are based on the CNN-based encoder-/decoder architecture U-Net (Fig 2), which has seen frequent use in the field of image-to-image transformation in medical imaging [26]. U-Net operates by encoding images into feature maps by concurrently reducing spatial resolution and increasing dimensionality, before decoding those feature maps into an image of the desired resolution while using skip-connections between encoder and decoder-blocks to preserve spatial information. For each up- and downsampling operation, the standard U-Net block comprises a 3x3 convolution, batch normalization, and rectified linear unit (ReLU)-activation (Fig 2b).

To improve upon the standard U-Net, we investigated a version where the standard block is replaced by one-shot aggregation (OSA)-blocks with efficient squeeze-and-excitation (eSE)-modules [27] (Fig 2d). OSA-blocks were proposed to optimize use of skip connections by reducing feature redundancy, in order to ultimately improve predictive performance while still being computationally efficient. eSE-modules are a development of squeeze-and-

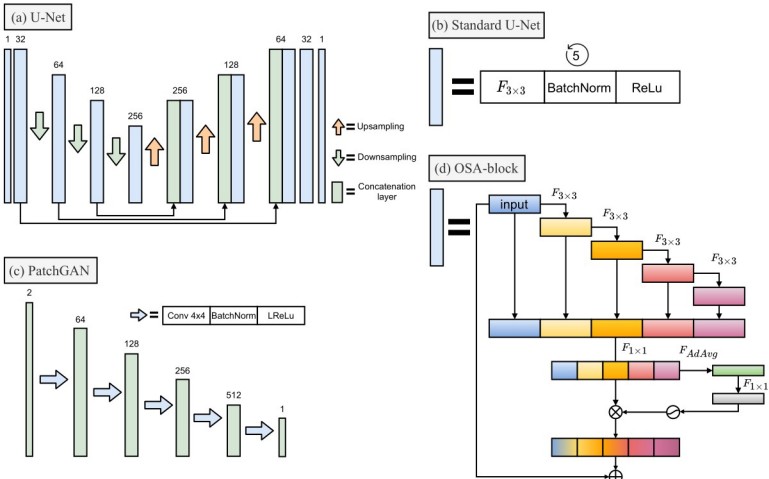

**Fig 2. Diagrams of ANN architectures used in this study.** (a) U-Net, a standard encoder/decoder architecture, which our image-to-image generator models are based on. Each upsampling uses bilinear interpolation to double image size and each downsampling uses max pooling to half image size. (b) The standard U-Net convolutional block, which contains a 3x3 convolution ($F_{3 \times 3}$), batch normalization (BatchNorm) and standard ReLu. (c) PatchGAN, which comprises the discriminator of our CGAN-based models. The two channels in the first PatchGAN-layer indicate the input z-sweep image stacked onto the corresponding z-projection, either predicted or measured. Each arrow indicates a convolutional block containing a convolution with a 4x4 kernel and stride of 2 (Conv 4x4), batch normalization (BatchNorm) and leaky ReLU activation function with alpha = 0.2 (LReLU). (d) One-shot aggregation (OSA) block, adapted from [27], which replaces the standard U-net block in our improved U-Net variants. Here, $F_{3 \times 3}$ and $F_{1 \times 1}$ indicate 3x3 and 1x1 convolutional layers, $F_{AdAvg}$ adaptive average pooling over spatial dimensions, $\otimes$ indicate element-wise multiplication and $\oplus$ element-wise addition. $F_{AdAvg}$ followed by the $1 \times 1$ convolution and the hard Sigmoid activation make up the efficient Squeeze-and-Excitation (eSE)-module of the OSA-block. Note that $F_{1 \times 1}$ projects the concatenated tensor to the same dimensions as input tensor and that the element-wise multiplication is performed along the channel-axis and broadcasted over the spatial dimension.

excitation (SE)-modules that were proposed to adaptively re-weight channels of encoded images in each convolutional block to further improve predictive performance [28].

To provide baseline generator models, we train both the standard U-Net and the OSA-U-- Net to minimize the L1-reconstruction error of the predicted z-projection compared to the ground truth. Formally, the L1 reconstruction error is given by

$$L_{L1}(P, \ \hat{P}) = \sum_{i=1}^{h}\sum_{j=1}^{w}|P_{ij} - \hat{P}_{ij}|. \tag{2}$$

That is, the L1-reconstruction error sums the absolute pixel-wise prediction errors.

## Conditional generative adversarial network

Whereas using purely reconstruction-based losses in an image-to-image application seems intuitive, in practice it often leads a network to favor proposals which are overly smooth or blurry. This is where a generative adversarial network (GAN)-based approach can offer improvement by applying two sub-models, a generator that proposes images and a discriminator to classify the proposal as real or fake, to drive proposals to become more realistic [29]. Based on the work of Isola et al., we investigated the use of conditional GANs (CGANs) to improve the generated z-projection images (summarized in Fig 3). In the CGAN approach, we have our generator, $G$, which is trained to predict the image $G(S) = \hat{P}$ based on $S$, where we wish $\hat{P}$ to be indistinguishable from the target $P$. We then add a secondary loss term to optimize $G$ to fool a discriminator model, $D$, which is trained to predict whether $P$ or $\hat{P}$, conditioned on the input $S$, is real or or not. That is, the discriminator is defined as $D: \mathbb{R}^{h \times w} \times \mathbb{R}^{h \times w} \rightarrow I^{\hat{h} \times \hat{w}}$, where $I = [0, 1]$ indicates the closed interval of predicted probabilities that each output node corresponds to a real pair and $\hat{h}$ and $\hat{w}$ are the output dimensions used in the CGAN PatchGAN-framework.

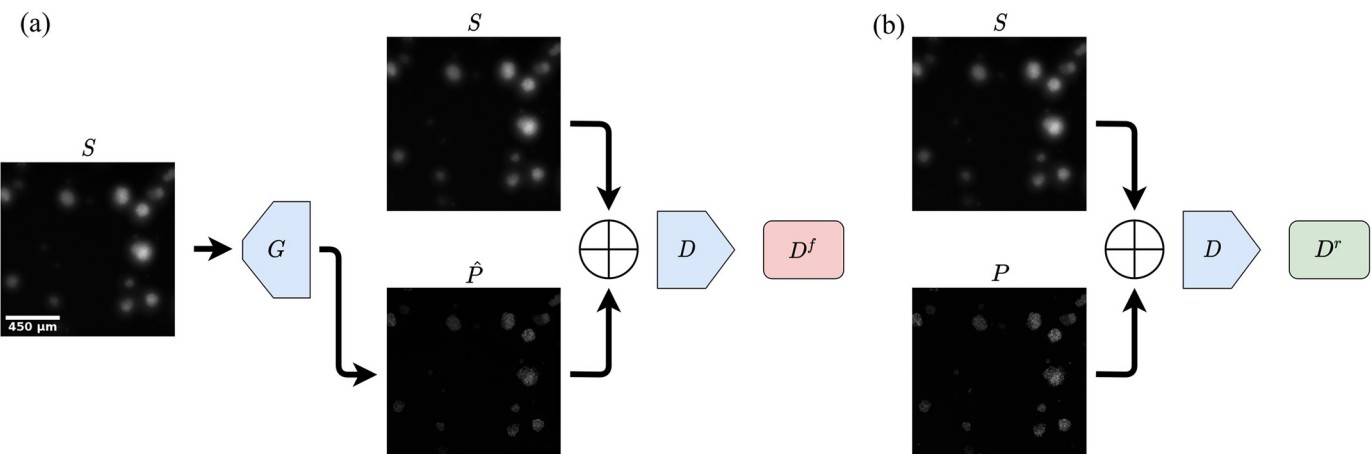

**Fig 3. Schematic illustration of the conditional generative adversarial network (CGAN)-approach used to improve prediction of z-projection of images.** $\oplus$ indicate stacking of the images. (a) During training, the generator $G$ predicts a z-projection image $\hat{P}$ based on the input sweep image $S$. $S$ is stacked onto $\hat{P}$ to create a two-channel image, which the discriminator $D$ attempts to classify as fake $D^f$. (b) To provide negative examples, real z-projection images $P$ are similarly stacked with z-sweeps and the discriminator then attempts to classify them as real $D^r$. The discriminator is optimized to accurately classify z-projection images as real or fake, while the generator network is optimized to make the discriminator prediction more difficult. The scale bar in the leftmost image applies to all images.

Formally, we optimize $G$ to minimize the weighted sum of the L1-loss (2) and an adversarial loss based on the discriminator prediction that is given by

$$L_G = \lambda_{L1} \; L_{L1}(P, \; \hat{P}) + \mathbb{E}[(D(S, \hat{P}) - R)^2] \qquad (3)$$

where $\lambda_{L1} \in \mathbb{R}$ is the weight of the L1-reconstruction loss, $R$ the $\hat{h} \times \hat{w}$ matrix of soft real labels, meaning that $R_{ij} \in [0.9, 1]$ for each $i \in [1, \hat{h}]$ and $j \in [1, \hat{w}]$ to use so called one-sided label smoothing, which Salimans et al. found to improve GAN-training [30]. The discriminator then minimizes the loss given by

$$L_D = \frac{1}{2}\mathbb{E}\big[(D(S, P) - R)^2 + (D(S, \hat{P}) - F)^2\big] \qquad (4)$$

where $F$ is the $\hat{h} \times \hat{w}$ matrix of fake labels, meaning that $F_{ij} = 0$ for each $i \in [1, \hat{h}]$ and $j \in [1, \hat{w}]$.

The CGAN-discriminator uses the PatchGAN architecture (Fig 2c), which has been found to improve texture quality of predictions [29]. Instead of the standard GAN-discriminator which performs binary classification of the input image as either real or fake [31], the Patch-GAN-discriminator can be viewed as a "sawed off" CNN, which outputs a 2D-tensor where the output neurons represent the discriminator classification of overlapping patches. To train the CGAN, the generator- and discriminator-weights are updated in an alternating fashion (illustrated in Fig 3). Discriminator weights are frozen during the first phase while the the generators weights are updated, optimizing the loss equation based on discriminator outputs (3). During the second phase, the generator weights are frozen to turn attention to the discriminator weights, which are updated to reduce loss defined in (4).

We investigated the use of two different generators for our CGAN architectures, an U-Net generator and an OSA-U-Net variant, both described in section **Image Generator**. The corresponding architectures are refereed to as CGAN (U-Net generator) and OSA-CGAN (OSA-U-Net generator) respectively and both versions use the PatchGAN discriminator described in the section above.

## Experiments

### Acquisition of datasets

To establish proof of concept across different kinds of biological specimens, we tested our proposed workflow on three common protocols for culturing live 3D tumor spheroids, each of which present different imaging challenges. Four common immortalized cancer cell lines were used across the experiments: A549 (lung cancer), MCF-7 (breast cancer), MDA-MB-231 (breast cancer) and SK-OV-3 (ovarian cancer). Commercially available cell lines stably expressing a nuclear-restricted mKate2 FP were used for A549 and MCF-7 (Sartorius), while standard SK-OV-3 (EACC) and MDA-MB-231 (ATCC) cell lines were purchased and stably transfected with the Incucyte® NucLight Red lentivirus reagent (EF1 Alpha Promoter; Sartorius) per manufacturer's instructions. Because the FP is nuclear restricted, it can be used to measure cell viability, i.e. a loss of viability results in a compromised nuclear membrane to cause a loss of fluorescence. The sections below describe the three protocols used to generate 3D tumor spheroid cultures.

**Single-spheroid preparation.** In this protocol, cancer cells are encouraged to form a single large tumor spheroid (Fig 4a), the size of which can present unique imaging challenges including limited light penetration and excess light scattering [32]. A549, MCF-7, MDA-MB-231 or SK-OV-3 cells were seeded into an ultra-low attachment round-bottom 96-well plate

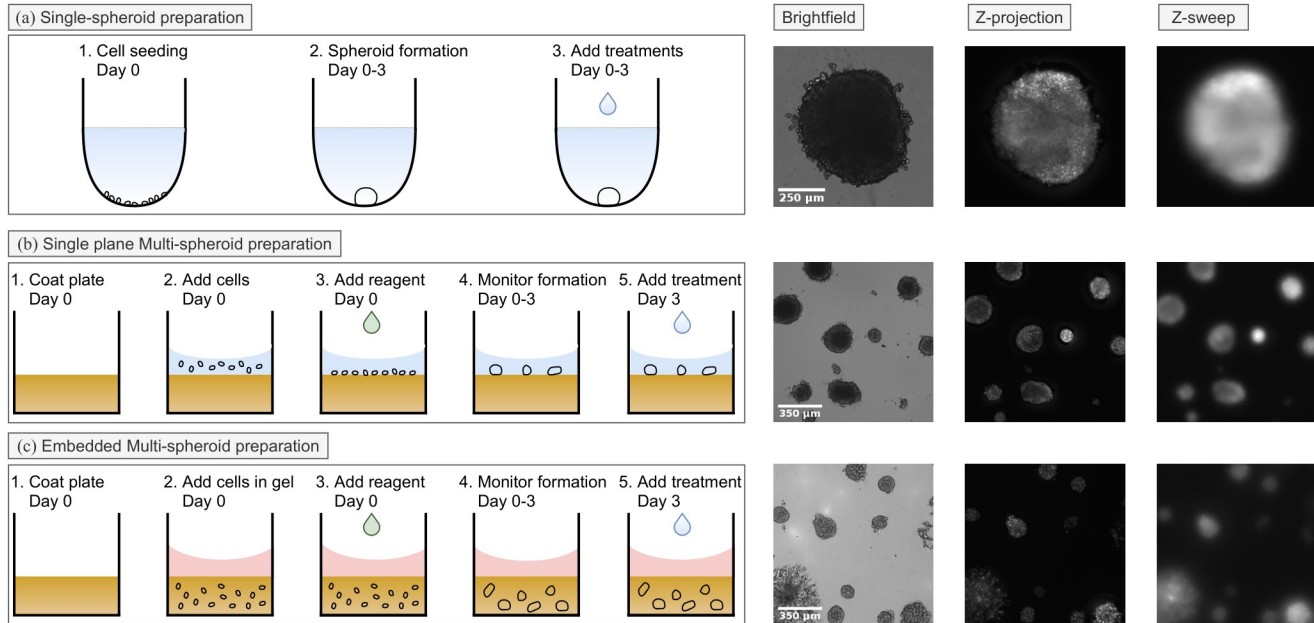

**Fig 4. Graphical summary of establishment and image acquisition of 3D tumour spheroid datasets.** (a) In a single spheroid assay, cells were seeded into a ultra-low attachment round bottom 96-well plate and allowed to form a single spheroid. (b) In a single plane multi spheroid assay format, cells were seeded into a polymerized bed of ®, where they grow into multiple spheroids in what is effectively a single optical plane. (c) In an embedded multispheroid assay, cells are resuspended in Matrigel® before placing in a flat bottom 96-well plate. Examples of brightfield, z-projection and z-sweep fluorescence images acquired 10 days post seeding are shown.

(Corning #7007), gently centrifuged at 150×g for 10min, and allowed to naturally form a single large spheroid of up to approximately 1mm in diameter over a 10 day period. At 3 days post seeding, 1$\mu$M of the cytotoxic compound camptothecin (CMP) was added to half of the wells. Spheroids continued to be monitored at regular intervals by the standard Incucyte® Spheroid module, before the custom fluorescence image acquisition was performed at 10 days post cell seeding once the control tumor spheroids had reached a size of approximately 1mm in diameter.

**Single-plane multi-spheroid preparation.** In this approach, rather than forming a single large spheroid, cancer cells are seeded on top of a flat extracellular matrix base to facilitate multiple tumor spheroid growth. Furthermore, tumor spheroids grow in what is effectively a single optical plane, simplifying matters for imaging (Fig 4b). To prepare for imaging, a flat-bottom 96-well plate (Corning #3595) was coated with 40$\mu$L of 4.5 mg/mL Matrigel® and polymerized at 37˚C for 30min. A549 or MCF-7 Cells were then seeded on top of the polymerized matrix and allowed to form spheroids. Multi spheroid formation was monitored by regular brightfield imaging with the standard Incucyte® Spheroid module, before the customized z-sweep and z-stack fluorescence image acquisition workflow was carried out at 10 days post cell seeding. Brightfield images were also collected in order to visualize the tumor spheroids.

**Embedded multi-spheroid preparation.** To visualize invasive cancer biology, an embedded multi-spheroid assay was prepared, which is the most complicated from an imaging perspective. Cancer cells are embedded within a volume of extracellular matrix, pushing the limits of the acquisition to resolve spheroids residing at varying z-locations (Fig 4c). Here, either MCF-7 or MDA-MB-231 cells were directly suspended in 50$\mu$L of 50% Matrigel®. The suspension was placed in each well of a cold 96-well flat bottom plate (Corning #3595), gently distributed across the entire bottom of the well and allowed to polymerize at 37˚C for 20min

resulting in a column of Matrigel® that is approximately 1.5mm tall. At 7 days post seeding, $1\mu$M CMP was added to half of the wells. In order to obtain kinetic data with which to validate our acquisition strategy, we performed the custom fluorescence image acquisition at 8, 9, 10 and 11 days post seeding.

## Image acquisition

Imaging was performed using customized acquisition software written to operate the motorized widefield fluorescence microscope within the Incucyte® S3 Live-cell imaging system (Sartorius). All images were collected using a 4X magnification objective (Navitar, NA = 0.2, FN = 26.5), CMOS camera (Basler Ace 1920-155um) and the Incucyte Red/Green optical module (Sartorius). While these experiments were performed on an Incucyte, where the camera moves in relation to a stationary sample, the acquisition strategy could presumably apply to any widefield fluorescent microscope with a motorized stage. For all experiments, the camera was moved to the center of the sample microwell and proceeded visualize fluorescence in the tumor spheroid using two image acquisition paradigms.

In the first paradigm, the traditional z-stack strategy was carried out. Here, a volume of either $750\mu$m (single spheroid or single-plane multi-spheroid) or 1.5mm (embedded multi-spheroid) was collected as a step-wise image series, where the camera repeats a plane-by-plane sequence of moving upward $25\mu$m along the z-axis starting from the bottom of the microwell, at each location pausing to acquire an image with a 50ms exposure time. Because the optimal Nyquist sampling rate is impractical and results in substantial photobleaching, the z-step distance was instead selected to be within the depth-of-field of the objective (DOF = $25.3\mu$m) in order to balance sampling rate and axial resolution. To minimize shot noise, we took 3 consecutive frames and averaged them together before moving to the next z-location. This sequence is then repeated throughout the entire volume. Each image series is then projected (Fig 1a, left) using a custom z-projection algorithm [33]. This custom projection scheme uses a neighborhood-based texture filter to determine the location in the z-stack in a manner similar to other texture-based projection algorithms [34], resulting in an image where the surface of each tumor spheroid object is in focus.

In the second acquisition paradigm, the camera covers the same axial distance as before by moving continuously along the z-axis at a speed of 1.25mm/s, with the goal of producing a single image where intensity is integrated across the z-dimension. While a single long exposure would have been preferred to accomplish this task, that risked saturating the 12-bit camera. Instead, the camera moved continuously while a consecutive series of 50 ms exposures with each exposure translating to motion across an approximately $65\mu$m distance. Consecutive exposures were summed into a single image to obtain the final z-sweep image (Fig 1a, right).

Both z-stack projections images and summed z-sweeps were normalized by their effective exposure time prior to neural network training, which was critical for maintaining the correlation of intensity values between the paired data to enable accurate neural network training. Care was also taken to minimize time between image acquisition paradigms for a given sample to maintain optimal data pairing for neural network training. In all cases, accompanying transmitted light brightfield image z-stacks were also collected in order to visualize the tumor spheroids.

## Data and training specifics

The single plane multi-spheroid dataset was used to train all models and consists of 96 different pairs of z-sweep and z-projection images. Out of these 96 images, 86 of them were chosen to be used for model training and the 10 remaining images were used to validate the models.

Data augmentation was used to enlarge the dataset, meaning that each pair of $1536 \times 1152$ images was cropped into multiple matching $256 \times 256$ sub-images at random locations. Cropped images were also flipped horizontally and vertically at random with a probability of 25% for both cases. Using data augmentation, the final training set consisted of 344 images and the validation set 40 images. All images were log-transformed during training. In comparison to batch-normalization, log-transformation keeps the relation of intensity between all the images the same and the network will therefore be able to learn to predict fluorescent intensity in a dynamic range. The images are available at figshare [35].

The generator models follow the U-Net and OSA-U-Net architectures described in section **Image Generator**. The CGAN-discriminator follows a PatchGAN-architecture, which consists of a CNN with basic convolutional blocks, batch normalization and leaky ReLu activations with alpha = 0.2, outputting 2D-tensors corresponding to $70 \times 70$ pixel overlapping receptive fields (see Fig 2c). To train the models, we used a mini-batch stochastic gradient decent configured with an Adam-optimizer [36], a learning rate of 0.0002 and batch size of 2. To train the CGAN-based models, the weight for the generator L1-loss, $\lambda_{L1}$, was set to 10. Also, one-sided label smoothing [30] was used for real images, meaning that a tensor of random values in the interval [0, 0.1] was subtracted from their target labels. Model check-pointing was used, meaning that all networks were trained for a maximal number of 1250 epochs and the models corresponding to the minimum validation error were used for further experiments. Specifically, U-Net was trained for 1027 epochs, OSA-U-Net for 1014, CGAN for 1217 and OSA-CGAN for 1227. For the CGAN-models, the validation error was given by the supervised L1-loss. The code can be found on GitHub [37].

## Results

### Model evaluation

To determine what type of CNN or method is most suitable for z-projection image reconstruction from z-sweep images, we compared four different models trained on the single-plane multi-spheroid dataset. All image generator models are based on U-Net [26], which is modified by adding adversarial training (CGAN) and/or replacing the standard convolutional blocks with OSA-blocks. To evaluate which model is able to predict the most accurate z-projection images from z-sweep images, we compare the image reconstruction performance using Frechet Inception Distance (FID) [38], Peak Signal to Noise Ratio (PSNR), Structural Similarity (SSIM), Multi Scale-SSIM (MS-SSIM) [39] and Mean Squared Error (MSE) on single-spheroid and single-plane multi-spheroid image pairs not used during training (see Fig 4a and 4c, and Table 1). These metrics all evaluate the perceived quality of the predicted image compared to a reference. While PSNR, SSIM and MSE are widely used in classical signal

**Table 1. Z-projection reconstruction performance on test set images as quantified by Frechet Inception Distance (FID), Peak Signal-to-Noise Ratio (PSNR), Structural Similarity (SSIM), Mean Squared Error (MSE) and Multiscale SSIM (MS-SSIM).** Performance is reported for the compared models as well as the input z-sweep to provide a baseline. Arrows indicate direction of improvement, meaning that an upward arrow indicates that a larger score is better and the opposite for a downward arrow.

| | Single-spheroid | | | | | Multi-spheroid | | | | |
|---|---|---|---|---|---|---|---|---|---|---|
| | FID (↓) | PSNR (↑) | SSIM (↑) | MS-SSIM (↑) | MSE (↓) | FID (↓) | PSNR (↑) | SSIM (↑) | MS-SSIM (↑) | MSE (↓) |
| Z-sweep | 119.61 | 18.85 ± 5.7 dB | 0.025 ± 0.016 | 0.986 ± 0.007 | 1886 | 294.25 | 17.97 ± 4.05 dB | 0.075 ± 0.045 | 0.936 ± 0.011 | 1831 |
| OSA-CGAN | 107.66 | 22.20 ± 9.19 dB | 0.023 ± 0.016 | 0.994 ± 0.005 | 3180 | 99.07 | 23.66 ± 4.8 dB | 0.107 ± 0.041 | 0.986 ± 0.009 | 550 |
| CGAN | 219.48 | 16.81 ± 12.03 dB | 0.015 ± 0.011 | 0.989 ± 0.010 | 249534 | 121.47 | 20.44 ± 5.74 dB | 0.089 ± 0.037 | 0.973 ± 0.012 | 1811 |
| OSA-U-Net | 134.84 | 22.40 ± 10.04 dB | 0.021 ± 0.015 | 0.993 ± 0.005 | 10609 | 113.59 | 22.45 ± 6.39 dB | 0.109 ± 0.041 | 0.985 ± 0.009 | 1189 |
| U-Net | 254.45 | 17.72 ± 11.37 dB | 0.015 ± 0.010 | 0.988 ± 0.010 | 10668 | 105.47 | 21.02 ± 5.64 dB | 0.081 ± 0.035 | 0.971 ± 0.008 | 1624 |

processing, FID was developed specifically to evaluate the quality of natural images generated by generative adversarial networks and MS-SSIM as an extension of SSIM with the purpose to asses the details of the image at multiple scales.

Out of the investigated models, OSA-CGAN performs best overall (Table 1). OSA-CGAN has the lowest FID, MSE, highest SSIM and MS-SSIM on both the multi- and single-spheroid datasets. The only metric OSA-CGAN is not the top model in is PSNR in the single-spheroid dataset where the OSA-U-Net performs slightly better. Indeed, we find that OSA-CGAN produces the images that are qualitatively the most similar to the z-projection ones (see Fig 5). We also find that both investigated modifications to U-Net, OSA-blocks and CGAN, contribute to the improved performance. In terms of FID on the single-spheroid data, both modifications improve relative to U-Net (FID = 138.84 for OSA-blocks and 219.48 for CGAN compared to 254.45 for U-Net) but less than them combined (FID = 107.66). While on the multi-spheroid images, each modification degrade performance (FID = 113.59 for OSA-blocks and 121.47 for CGAN compared to 105.47 for U-Net) but improve performance when combined (FID = 99.07). We draw similar conclusions from PSNR, SSIM, MS-SSIM and MSE. Even though MSE on the single-spheroid images for CGAN is a heavy outlier. Qualitatively, we find that both CGAN and OSA-blocks improve the sharpness of the generated images, however not

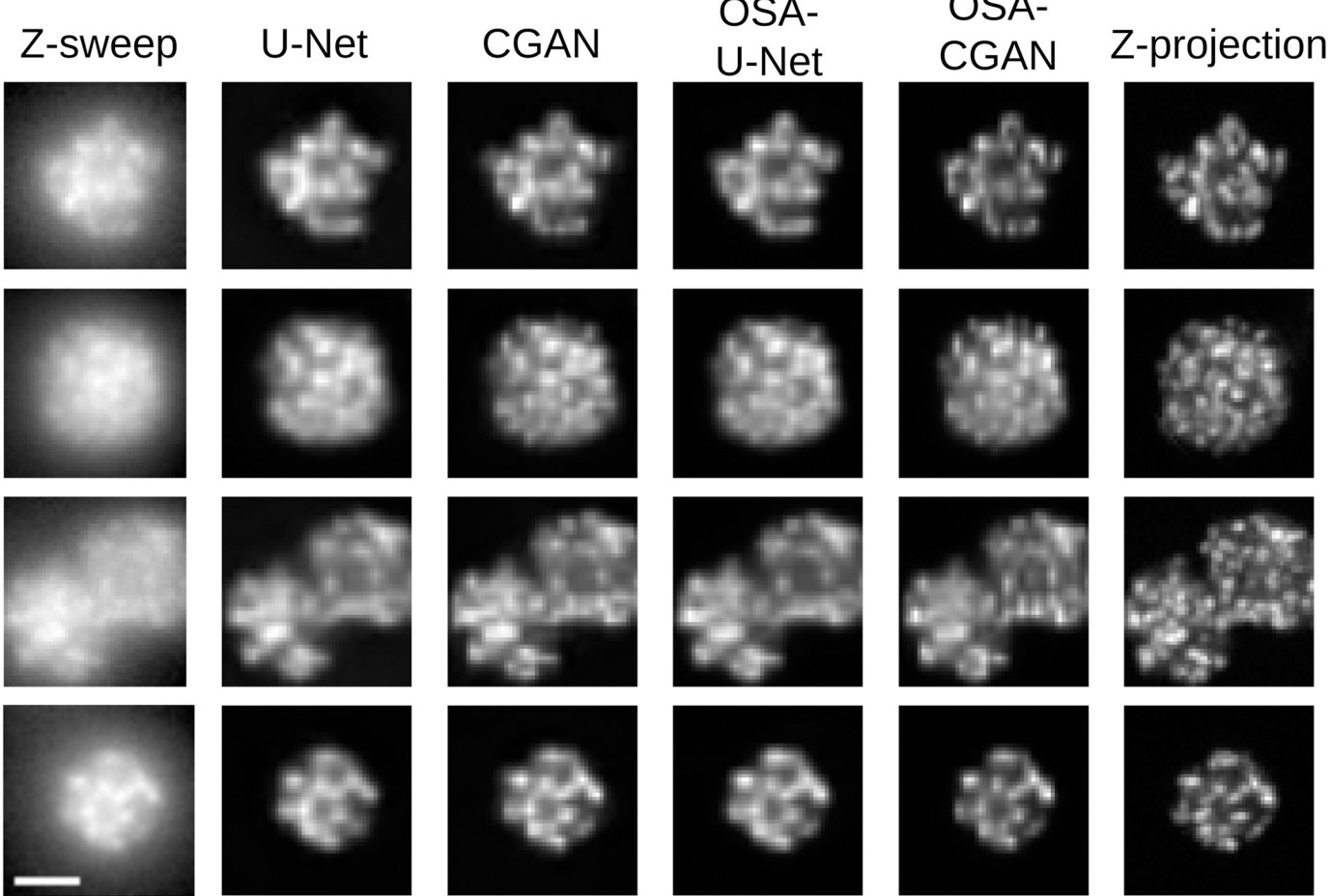

**Fig 5. Illustrative examples of z-sweeps, z-projections and model predicted reconstructions for the compared models.** The scale bar in the bottom left image is 50 $\mu m$ and applies to all images. The brightness of the images are relative for visual purposes.

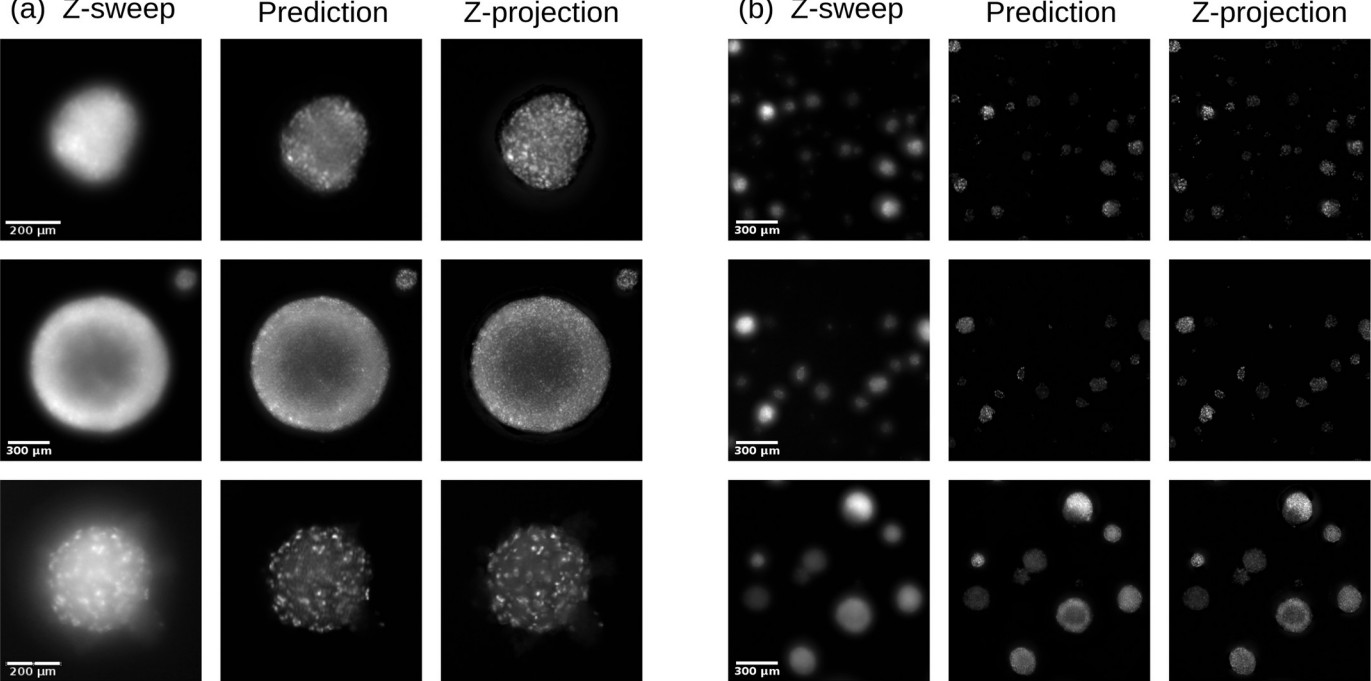

**Fig 6. Qualitative reconstruction performance of OSA-CGAN.** On single-spheroid images (a) and multi-spheroid ones (b). The brightness of the images are relative for visual purposes.

over the full dynamic range of the images and appear comparatively blurrier to the z-projection. Our proposed OSA-CGAN on the other hand, generates perceptually sharp images over the full dynamic range (Figs 5 and 6).

## Fluorescence fidelity

A common use case of 3D fluorescent imaging of tumor spheroids is to quantify the mean fluorescence intensity within the spheroid to measure biological response, e.g., cell health and viability. In order to use predicted z-projections in such applications, it is critical to not only predict images that are perceptually similar, but they also need to carry the same information without introducing systematic bias. To assess whether our proposed OSA-CGAN allows for the same spheroid-wise quantification as the z-projection, we detect individual spheroids in the multi-spheroid dataset and quantify the spheroid-wise logged sum of fluorescence intensities and compare to z-projection-based quantification (Fig 7).

We find that OSA-CGAN allows for equivalent quantification as the z-projection without introducing systematic bias ($R^2$ = 99.4%, slope = 1, Fig 7c). In comparison, the z-sweep intensities are noisier compared to the predicted z-projections and systematically underestimates the intensity. We conclude that OSA-CGAN faithfully reproduces the z-projection fluorescence intensities on an individual tumor spheroid basis.

## Fluorescence-based measure of cell death in embedded spheroids

To demonstrate that our workflow allows equivalent interpretation in biologically relevant datasets as traditional fluorescence microscopy, we investigated fluorescence intensity-based measurements from the different acquisition paradigms to quantify loss of cell viability over

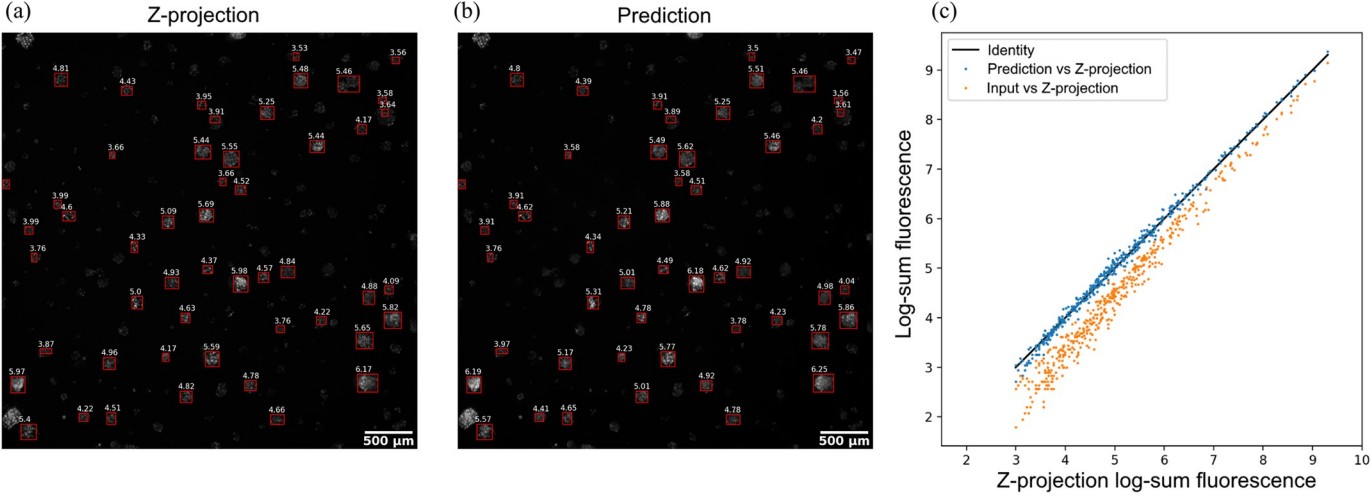

**Fig 7. Comparison of mean fluorescence intensity measurements of tumor spheroids shows fidelity between ground true and predicted z-projection images.** Representative example of (a) true z-projection and (b) OSA-CGAN predicted z-projection image, with mean intensity values overlaid for each tumor spheroid. (c) Logscale scatterplot of mean intensity values of predicted vs. ground truth (blue) and predicted vs. input z-sweep images (orange).

time. Taking our best performing model from above, we applied the OSA-CGAN to z-sweeps of a dataset comprising embedded MDA-MB-237 and MCF-7 tumor spheroids, which were either vehicle or CMP-treated to induce cell death. Here, because tumor spheroids are embedded throughout the Matrigel®, spheroids reside at different locations along the z-dimension. To quantify fluorescence, we segmented the brightfield images of the tumor spheroids using a custom version of the Incucyte® Spheroid Analysis module. Using those brightfield segmentation masks, we quantified the average mean fluorescence intensity of tumor spheroids in each image to compare the raw z-sweep, OSA-CGAN prediction and z-projection images. Masked objects with area less than 1200 $\mu$m$^2$ were not included in the analysis, in order to exclude debris or other brightfield imaging artifacts. Results are shown in Fig 8.

Using this strategy, we find that the predictions from our proposed method correspond very well to the gold standard z-projection (Fig 8b compared to Fig 8c). Both display similar

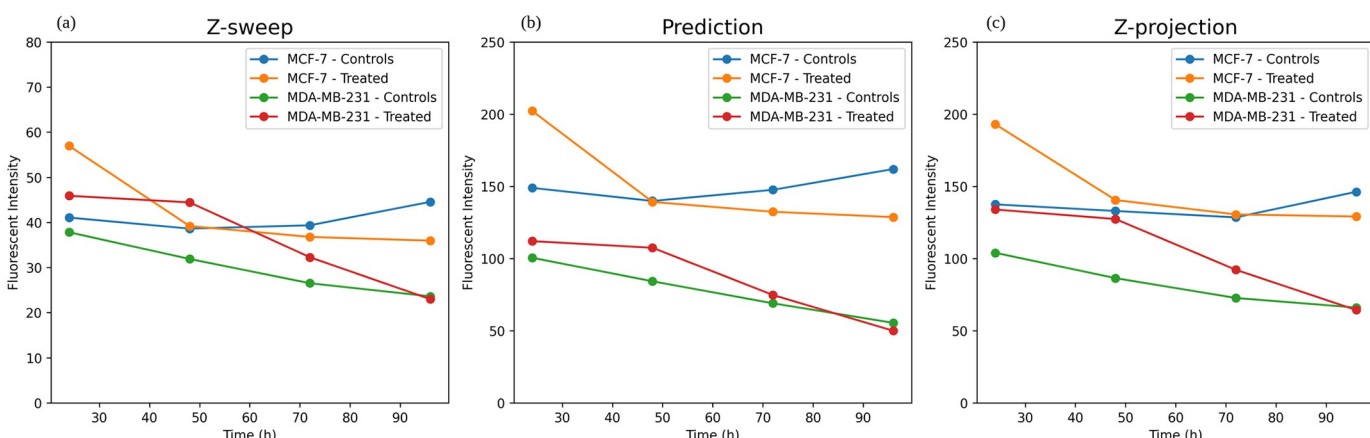

**Fig 8. Time courses of nuclear-restricted mKate2 fluorescence intensities of embedded tumor spheroids.** Quantified based on z-sweeps (a), OSA-CGAN predictions (b), and z-projections (c). Note the different scale of y-axis in A.

trends over time, both in terms of absolute intensities and relative changes, to show the loss in cell viability in response to CMP treatment compared to the control. In comparison, the quantification based on the raw z-sweeps are underestimated relative to the z-projection (Fig 8a). Although the general trends are similar, the z-sweep quantification shows that early time-point intensities of treated MDA-MB-237 are higher in terms of absolute intensities. For example, at the first time point, the CMP-treated MDA-MB-237 exhibit higher intensity than the vehicle-treated MCF-7 at the first time point and is higher than both MCF-7 cultures at the second time point. This relationship is not observed in the quantification based on z-projection images nor predicted images. This suggests that there is potential for mischaracterizing fluorescence-based data of the raw z-sweep acquisition, which can actually be corrected by our trained OSA-CGAN model.

## Discussion

Widefield fluorescence imaging has been an indispensable tool for advancement of biological research for decades. However, experienced users are all too familiar with the trade-offs that need to be considered when imaging a 3D sample, namely balancing throughput and minimal risk of phototoxicity and photobleaching with maximal signal-to-noise and axial resolution [2]. Here, we present an approach to circumvent those trade-offs without the need of additional hardware, by coupling deep learning-based image restoration with an unconventional acquisition paradigm to achieve high-quality, quantitative 2D projection images of 3D samples. This proposed method enables high-throughput widefield fluorescence imaging in a manner that has not previously been feasible for large 3D samples, permitting lines of investigation that have previously only been possible in 2D models or with specialized optical setups. For example, efficient anti-cancer compound screening, where fluorescence data is typically used to quantify cell health and death, requires many different drugs and concentrations to be tested simultaneously. Because such high-throughput is a requirement for these experiments, researchers either use 2D cell culture models, which are not necessarily relevant to true in vivo conditions, or they perform very limited axial sampling of their 3D samples, which risks biasing or leaving behind data. Not only does the proposed method capture fluorescent images in a fast, high-throughput manner to generate qualitatively impressive images (Fig 6), but it also maintains the fidelity of quantitative measurements (Fig 7) making it suitable for measuring biological response, e.g., cell death in response to drug treatment (Fig 8). Notably, for our embedded tumor multi-spheroid samples, which required acquisition of data across a 1.5mm volume, the conventional z-stack approach took 101 seconds for a single sample whereas the z-sweep only took 1.2 seconds, nearly achieving a 100-fold speedup. This demonstrates our proposed method's utility for high-throughput quantitative fluorescent imaging of large 3D samples, which enables the same data measurements in a fraction of the time of the conventional z-stack approach. Additionally, our approach does not require sophisticated hardware to generate images quickly once the model has been trained. A trained OSA-CGAN generates one image in 0.33 seconds on a laptop equipped with an NVIDIA Quadro RTX 3000 and requires 4015MiB of VRAM. Running the same model on an Intel Core i7-CPU @ 2.30GHz takes 10 seconds per image. This indicates that our workflow could lend itself quite well to high-throughput, fast high-quality projection imaging of large 3D sample.

Worth noting is that the z-stack approach sometimes suffers from halo-like artefacts surrounding the cell clusters, typically due to light scattering typical to widefield fluorescence [3]. This artefact is apparent with close inspection of the ground truth images in Fig 6. Because these artefacts are present in the training dataset, one could expect these to negatively affect the model prediction; however, this is not the case. Instead, the model learns to ignore these in

favor of the biologically relevant information present in the images, resulting in predictions that are free of artefacts. This phenonemon has been observed in other applications such as optoacoustic image reconstruction from sparse data [40]. While the underlying reason is not yet understood, we speculate that the artefacts may not occur consistently enough to provide a strong signal compared to the signal of interest. Unless the model is overfitted, it may then not learn to reproduce the z-projection images completely according to the training data and may hence not include these artefacts. To investigate the reason for how CNN-based image reconstruction may provide implicit artefact removal is something we leave for future research.

Although we compare two different CNN-architectures, specifically showing that a modified U-Net and using OSA-blocks is preferable to a standard U-Net generator, we do no extensive study of how architectural elements affect the result. CNN-architecture design is a large field of study and there are vast permutations and options for how to design the CNN optimally. While we chose OSA-blocks, we could have obtained similar results by choosing other architectural variants that have demonstrated better performance than the straightforward fully convolutional blocks used in the standard U-Net (e.g., split-attention blocks [41] or dense blocks [42]). Similarly, many variants of generative adversarial networks have been described in recent literature. We chose the conditional GAN variant due to its extensive track record of successful applications, including remote sensing image analysis [43], stain-to-stain-translation in histopathological images [44], translating between different type of magnetic resonance imaging-weighting [45], and more. Regardless of the neural network architecture used, images that result from a z-sweep acquisition are well-suited for a deep learning based restoration workflow. While indeed appearing low in quality, z-sweep images effectively represent the fluorescent intensity of the sample integrated over the z-axis. Therefore, we speculate that the z-sweep sufficiently provides a neural network with enough real information about the sample to accurately predict the restored unintegrated fluorescent signal. The main focus of this study is to demonstrate the combination of that axial z-sweep acquisition and deep learning-based image restoration and we leave an extensive evaluation of CNN-architectures and training methodologies for future research.

Our deep learning models were solely trained on single-plane multi-spheroid data and managed to generated good results on data from all three experimental settings, exhibiting model transferability across plates. We note that the SSIM and MSE score when evaluating on single-spheroid data was better for the input Z-sweep images than the deep-learning model projections (Table 1). The FID metrics, MS-SSIM and PSNR scores on the other hand showed an improvement for the projections which is in line with can be visibly be observed (Fig 6a). However, there are certain caveats related to the z-sweep images that are important to consider in relation to these metrics. Because the z-sweep has a much longer effective exposure time than the z-stack projections, z-sweeps exhibit less shot noise. However, because the z-sweep acquisition moves through the sample so quickly, it also captures less photons at a given z-location resulting in the overall signal being lower. Applying this to our different biological models, the large, very bright single spheroid model, metrics are likely less affected by the signal difference in the acquisition and more impacted by shot noise. Whereas in the multi-spheroid models, the smaller, dimmer spheroids bias the metrics toward changes in signal. Because of these caveats, we feel that the FID score, which is affected by noise and blurriness but less so overall intensity, is the most meaningful of the metrics investigated.

While our results detail an important step to enable this workflow for high-throughput fluorescence imaging of large 3D samples, there are certain limitations. Here, we have tested three common different experimental preparations for culturing tumor spheroids, which range from a single, relatively large spheroid to hundreds of spheroids embedded throughout a volume of extracellular matrix (Fig 4). While this demonstrates the applicability of our

proposed method to a variety of 3D samples, the world of 3D imaging in biology is incredibly diverse. Future investigations will certainly focus on validating this approach on additional 3D models, such as organoids or developmental biology model organisms, as well as model generalizability to other target proteins and FPs, particularly in relation to FP wavelength and brightness. Furthermore, expanding this workflow to other fluorescence modalities, such as lattice light sheet and spinning disk confocal, will also be a promising line of future investigation. The z-sweep acquisition strategy has already been demonstrated to be compatible with lattice light sheet [10], so we speculate our deep learning-based image enhancement workflow will readily adapt to other modalities.

## Conclusions

Widefield fluorescence microscopy is commonly used to study biological phenomena. However, fluorescent microscopic imaging of complex 3D samples, such as tumor spheroids, is burdensome since it often entails the collection of a stack of multiple optical sections along the z-dimension, which is time-consuming and has high risk of causing phototoxicity. To alleviate this problem, we propose a workflow combining axial z-sweep acquisition and deep learning-based image enhancement. Axial z-sweep acquisition is many times faster than the corresponding z-stack acquisition but suffer from poor image quality. Using deep convolutional neural networks and conditional generative adversarial networks, we train models to predict high quality projection images of z-stacks based on z-sweep images. We demonstrate that our proposed model, OSA-CGAN, outperforms baseline models in terms of visual quality, quantitatively and qualitatively, in three case studies of tumor spheroid imaging. OSA-CGAN is able to produce perceptually sharp images without introducing systematic bias in fluorescent imaging of single-, multi- and embedded tumor spheroids. To conclude, this work shows how axial z-sweep acquisition and deep learning enable high-throughput and high-quality widefield fluorescence microscopy imaging of complex 3D samples.

## Author Contributions

**Conceptualization:** Christoffer Edlund, Rickard Sjögren.

**Data curation:** Miniver Oliver, Kalpana Barnes.

**Formal analysis:** Edvin Forsgren.

**Investigation:** Edvin Forsgren.

**Methodology:** Edvin Forsgren, Christoffer Edlund, Timothy R. Jackson.

**Project administration:** Timothy R. Jackson.

**Resources:** Miniver Oliver, Kalpana Barnes.

**Software:** Edvin Forsgren, Christoffer Edlund.

**Supervision:** Christoffer Edlund, Rickard Sjögren, Timothy R. Jackson.

**Validation:** Edvin Forsgren, Timothy R. Jackson.

**Visualization:** Edvin Forsgren.

**Writing – original draft:** Edvin Forsgren, Christoffer Edlund, Rickard Sjögren, Timothy R. Jackson.

**Writing – review & editing:** Edvin Forsgren, Christoffer Edlund, Rickard Sjögren, Timothy R. Jackson.

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
