## [Decision Letter · Decision Letter 0]

14 Mar 2022

PONE-D-22-03425High-throughput widefield fluorescence imaging of 3D samples using deep learning for 2D projection image restorationPLOS ONE

Dear Dr. Forsgren,

Thank you for submitting your manuscript to PLOS ONE. After careful consideration, we feel that it has merit but does not fully meet PLOS ONE’s publication criteria as it currently stands. Therefore, we invite you to submit a revised version of the manuscript that addresses the points raised during the review process.

We look forward to receiving your revised manuscript.

Kind regards,

Sathishkumar V E

Academic Editor

PLOS ONE

Journal Requirements:

2. In your Methods section, please include information on the source or supplier of the cell lines used.

Reviewers' comments:

Reviewer's Responses to Questions

**Comments to the Author**

1. Is the manuscript technically sound, and do the data support the conclusions?

Reviewer #1: Yes

Reviewer #2: Partly

2. Has the statistical analysis been performed appropriately and rigorously? 

Reviewer #1: Yes

Reviewer #2: No

3. Have the authors made all data underlying the findings in their manuscript fully available?

Reviewer #1: No

Reviewer #2: No

4. Is the manuscript presented in an intelligible fashion and written in standard English?

Reviewer #1: Yes

Reviewer #2: No

5. Review Comments to the Author

Reviewer #1: 1. Comparison study is not provided.

2. As you mentioned that 96 images were used for research. It is not available in public.

3. The deep learning model techniques, results need to compare.

4. Provide in detail about OSA-CGAN.

5. Overall the idea was good.

Reviewer #2: Widefield fluorescence microscopy is commonly used to study biological phenomena. However, fluorescent microscopic imaging of complex 3D samples, such as tumor spheroids, is burdensome since it often entails the collection of a stack of multiple optical sections along the z-dimension, which is time-consuming and has high risk of causing phototoxicity. To alleviate this problem, we propose a workflow combining axial z-sweep acquisition and deep learning-based image enhancement. This paper is novel and the contributions are good for a journal article. The revised version of the paper may be considered for publication in this journal.

1. The literature of the paper is poor, the authors may be consider the most recent articles for literature.

2. The reasons to achieve the superior performance of the article may be included in the revised version of the article.

3. List the limitations of the proposed work.

4. Discuss the complexity of the proposed model and how efficient it is over the existing ones.

5. Provide the dataset details or citations.

6. There are several performance metrics in the literature, whereas the authors consider only few in the paper.

6. PLOS authors have the option to publish the peer review history of their article (what does this mean?). If published, this will include your full peer review and any attached files.

Reviewer #1: **Yes: **ANANDHAN K

Reviewer #2: No

---

## [Author Response · Author response to Decision Letter 0]

27 Apr 2022

Please see the Response to reviewers file.

---

## [Decision Letter · Decision Letter 1]

6 May 2022

High-throughput widefield fluorescence imaging of 3D samples using deep learning for 2D projection image restoration

PONE-D-22-03425R1

Dear Dr. Forsgren,

We’re pleased to inform you that your manuscript has been judged scientifically suitable for publication and will be formally accepted for publication once it meets all outstanding technical requirements.

Kind regards,

Sathishkumar V E

Academic Editor

PLOS ONE

Additional Editor Comments (optional):

Reviewers' comments:

Reviewer's Responses to Questions

**Comments to the Author**

1. If the authors have adequately addressed your comments raised in a previous round of review and you feel that this manuscript is now acceptable for publication, you may indicate that here to bypass the “Comments to the Author” section, enter your conflict of interest statement in the “Confidential to Editor” section, and submit your "Accept" recommendation.

Reviewer #1: All comments have been addressed

Reviewer #2: All comments have been addressed

2. Is the manuscript technically sound, and do the data support the conclusions?

Reviewer #1: Yes

Reviewer #2: Yes

3. Has the statistical analysis been performed appropriately and rigorously? 

Reviewer #1: Yes

Reviewer #2: Yes

4. Have the authors made all data underlying the findings in their manuscript fully available?

Reviewer #1: Yes

Reviewer #2: No

5. Is the manuscript presented in an intelligible fashion and written in standard English?

Reviewer #1: Yes

Reviewer #2: Yes

6. Review Comments to the Author

Reviewer #1: The author has made the dataset public. The author addressed all the reviewer queries properly. All the performance evaluations are provided in the table. I don't have any objection to proceeding with the publication process.

Reviewer #2: The authors addressed all the recommended comments and the current version is recommended to publish in this journal. Congratulations to the authors.

7. PLOS authors have the option to publish the peer review history of their article (what does this mean?). If published, this will include your full peer review and any attached files.

Reviewer #1: **Yes: **ANANDHAN K

Reviewer #2: No

---

## [Editor Report · Acceptance letter]

10 May 2022

PONE-D-22-03425R1 

High-throughput widefield fluorescence imaging of 3D samples using deep learning for 2D projection image restoration 

Dear Dr. Forsgren:

I'm pleased to inform you that your manuscript has been deemed suitable for publication in PLOS ONE. Congratulations! Your manuscript is now with our production department. 

Kind regards, 

on behalf of

Dr. Sathishkumar V E 

Academic Editor

PLOS ONE